# The methods of improving cultural sensitivity of depression scales for use among global indigenous populations: a systematic scoping review

## Overview Review

psychometrics; depression; Indigenous; culturally competent care; cross-cultural psychiatry

**Corresponding author:**
Outi Linnaranta;
Email: outi.linnaranta@thl.fi

L.G.C. and M.Y. shared co-first authorship.

Liliana Gomez Cardona[1,2], Michelle Yang[3], Quinta Seon[1,2], Maharshee Karia[1], Gajanan Velupillai[2], Valérie Noel[1,4] and Outi Linnaranta[1,2,5]

[1]Douglas Mental Health University Institute, McGill University, Montreal, QC, Canada; [2]Department of Psychiatry, McGill University, Montreal, QC, Canada; [3]École interdisciplinaire des sciences de la santé/Interdisciplinary School of Health Sciences, Université d'Ottawa/University of Ottawa, Ottawa, ON, Canada; [4]ACCESS Open Minds, Douglas Mental Health University Institute, McGill University, Montreal, QC, Canada and [5]Equality Unit, Finnish Institute for Health and Welfare, Helsinki, Finland

## Abstract

Cultural adaptation of psychometric measures has become a process aimed at increasing acceptance, reliability, and validity among specific Indigenous populations. We present a systematic scoping review to: (1) identify the depression scales that have been culturally adapted for use among Indigenous populations worldwide, (2) globally report on the methods used in the cultural adaptation of those scales, and (3) describe the main features of those cultural adaptation methods. We included articles published from inception to April 2021, including 3 levels of search terms: Psychometrics, Indigenous, and Depression. The search was carried out in the Ovid Medline, PubMed, Embase, Global Health, PsycINFO, and CINAHL databases, following PRISMA guidelines. We identified 34 reports on processes of cultural adaptation that met the criteria. The scales were adapted for use among Indigenous populations from Africa, Australia, Asia, North America, and Latin America. The most common scales that underwent adaptation were the Patient Health Questionnaire (PHQ-9), the Center for Epidemiologic Studies Depression Scale (CES-D), and the Edinburgh Postnatal Depression Scale (EPDS). Methods of adaptation involved a revision of the measures' cultural appropriateness, standard/transcultural translation, revision of the administration process, and inclusion of visual supports. Culturally safe administration of scales was reported in some studies. To come to a consensus on most appropriate methods of improving cultural safety of psychometric measurement, most studies utilized qualitative methods or mixed methods to understand the specific community's needs. Revision of linguistic equivalence and cultural relevance of content, culturally safe administration procedures, qualitative methods, and participatory research were key features of developing safe culturally adapted measures for depressive symptoms among Indigenous populations. While for comparability, uniform scales would be ideal as mental health evaluations, an understanding of the cultural impact of measurements and local depression expressions would benefit the process of developing culturally sensitive psychometric scales. PROSPERO registration ID: CRD42023391439.

## Impact statement

Language, context, and methods of administrating psychiatric evaluations may hinder timely and accurate depression evaluation of Indigenous peoples, which may delay or prevent treatment. A potential strategy to close this gap is to develop new or culturally adapted scales that are more sensitive to detecting culturally specific expressions of mental health. While several structured instruments exist to describe methods and phases of adaptation, it remains open how commonly methods of adaptation are used and reported. In this research, we review original work representing all continents and report on the global evidence on the types of adapted scales for use with Indigenous peoples, their qualities, and their methods of adaptation. This is hoped to form a basis for future work in identifying suitable scales for adaptation, and in developing, adapting, and creating culturally sensitive psychometrics for treatment of major depression disorder. Given the need for equal access to safe mental health services for all population groups, it remains to be defined when a culturally safe administration of a standard scale is preferable, when there is a need to invest in a translation or a cultural adaptation of a scale, and to what extent visual supports can complement verbal measurement. This summary of findings on successfully adapted scales as well as acceptance of items in specific scales can support clinicians who need to choose a safe measure for assessing patients from different cultures. Moreover, the importance of inclusion of community members in the process of selecting and adapting the psychometric scales should be recognized in research and clinical work.



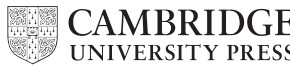

## Introduction

Indigenous peoples have originally inhabited several geographical regions and formed values, languages, and strong links to these territories. Indigenous people number about 400 million worldwide, belonging to five thousand diverse groups living in more than 90 countries (Indigenous Peoples, 2023; United Nations, 2023). As a result of imperialism and colonialism, the health of Indigenous peoples worldwide is characterized by systematic inequities in health outcomes (Reid et al., 2019). They have experienced varying degrees of racial discrimination and social oppression, evident as health and social marginalization in regard to education, housing, employment, and access to adequate social and health services (Frohlich et al., 2006). It is an alarming fact that such health disparities are still predominant to this day. This is well illustrated by the higher prevalence and morbidity/mortality rate of mental health conditions and suicide among Indigenous people, up to 20 times higher than the general population (Hajizadeh et al., 2019). This emphasizes the need for providing efficient and accessible mental health care, among other procedures, to improve Indigenous populations' resilience (Davy et al., 2016), and calls for a decolonization of psychological practices (Smallwood et al., 2023). Ensuring the equal accessibility of mental health services requires more than easy reach services; it requires examination of barriers and racist and colonial practices and then implementation of culturally appropriate mental health services (Davy et al., 2016; National Collaborating Centre for Aboriginal Health, 2019).

Accessibility of medical care is traditionally dependent on formal diagnoses by conventional psychiatric professionals. Yet, the traumatic history of colonization has negatively impacted Indigenous populations' help seeking behavior, trust, and acceptance of Western psychiatric treatment (Chachamovich et al., 2015). Unfortunately, delayed efforts and unfruitful cooperation are common in screening and treating mental health (Leung, 2016). Furthermore, Indigenous people share values and historical experiences that influence their expression of symptoms of depression – cultural expressions not represented in the biomedical conceptualization of psychiatric disorders (Kleinman, 1977; Kirmayer et al., 2008; Nichter, 2010). Use of colonializers' language and structural barriers have hindered access to good quality mental health and social services (Montreal Urban Aboriginal Health Needs Assessment, 2012). Without adequate levels of cultural safety embedded in current psychiatric practice (e.g., cultural awareness, cultural sensitivity, and cultural competency), there is limited timely and effective response to depression among the Indigenous (Brascoupé and Waters, 2009; Darroch et al., 2017).

Access to psychiatric care is based on diagnostic categories; for example, severe depression is characterized by the symptoms that decrease the ability to function (Aboraya et al., 2018). The 'gold-standard' method of psychiatric diagnosis is clinical interviews, where the delivery is through human interaction. However, Indigenous communities, especially remote groups, typically have limited access to personnel with specialized skills (Boksa et al., 2015). To overcome this barrier, screening for psychiatric disorders can be completed through psychometric measures in primary care to guide decision-making in treatment. In instances where psychiatric services are not easily accessible, psychometric measures may be used during follow-ups to detect relapse, and inform self-management and therapeutic interventions (Boksa et al., 2015). The organizational and political need for accurate measurement of a population's mental health increases when there is a need to allocate resources, plan service, or to improve quality of treatment.

If initial contact for mental health evaluation and treatment includes completing a psychometric screen, it is essential to confirm that this method of evaluation is culturally acceptable. Currently, most available instruments are based on the Diagnostic and Statistical Manual-5 (DSM-5), which was developed based on the literature and contribution of subject experts predominantly from Western societies. Additionally, as the DSM-5 was developed for use in the context of Western psychiatric practices, it does not capture the global expression of mental health conditions (Kleinman, 1977; Haroz et al., 2017). Simultaneously, calls for cultural safety in health care education and services and concurrent adaptation of measures for assessing psychiatric symptoms in Indigenous populations have been made (Brascoupé and Waters, 2009; Baba, 2013; Darroch et al., 2017).

The cultural adaptation of assessment tools has improved acceptance, response rate, and reliability of measures (Arafat et al., 2016; Kral, 2016; Hackett et al., 2019). There has been work on guidelines for the process of cross-cultural adaptations (Beaton et al., 2000; Chávez and Canino, 2005; Sousa and Rojjanasrirat, 2011; Arafat et al., 2016). This process generally involves modifications of instruments with the aim of making them more suitable for a specific population (Wexler et al., 2017; Haroz et al., 2017; Wiltsey Stirman et al., 2019). However, global consensus on guidelines and evaluation criteria for cultural adaptation is not available, and actually used methods remain open.

In this paper, we present a scoping review to synthesize the literature on cultural adaptations of depression measures among Indigenous populations. The rationale of this study is based specifically on a problem identified during the course of our participatory research with Indigenous populations – a critical Indigenous view on conventional, symptom-focused psychometrics (Gomez Cardona et al., 2021). The main objectives of this review were to: (1) identify the depression scales that have been culturally adapted for use among Indigenous populations worldwide, (2) report the methods used in the cultural adaptation of those scales, and (3) describe the main features of used cultural adaptation methods.

## Methods

### Study design

We conducted a systematic scoping review, in accordance with the Preferred Reporting Items for Systematic Reviews and Meta-analyses extension for Scoping Reviews (PRISMA-ScR) guidelines for scoping reviews (Tricco et al., 2018). The PRISMA-ScR checklist provides a systematic process for reporting title, abstract, introduction, methods, results, discussion, and funding (Tricco et al., 2018). A systematic scoping review design was chosen as this allowed us to present an overview of a large and diverse body of literature pertaining to the broad topic of screening for Indigenous mental health, while carrying out a systematic approach to answering focused objectives.

### Search strategy

We used three levels of search terms, which included terminology for depression, psychometrics, and Indigenous (Supplement 1). We searched Ovid Medline, PubMed, PsycINFO, Embase, CINAHL,

and Global Health databases for original research articles published between inception of the databases to April 2021. Gray literature searches were complemented through hand-searching on Google Scholar, open access repositories and the reference lists of selected articles. Study researchers (MY, QS) and a librarian created the search strategy and conducted the search. Citations of the search results were uploaded to a citation manager, and the library of references was accessible to all co-authors. The original search was conducted in July 2019, and supplemented by an updated systematic search in April 2021, under the guidance of a McGill University librarian.

### Operationalization of search terms

We operationalized the term *Indigenous* as the original habitants of colonized geographical regions (e.g., European colonization) who have cultural characteristics (e.g., values, languages, strong links to territories) distinct from the dominant societies, self-identify as Indigenous/Aboriginal peoples, and live in their country of origin (United Nations, 2023). However, where possible and to be respectful, we use the terms specified by the specific Nation. We considered cultural adaptation of scales in the form of questionnaires, surveys, or self-reports in culturally distinct populations. *Cultural adaptation* was operationalized as modifications to increase cultural sensitivity, not solely through – but including – the use of translation. *Depression* was operationalized in this study as psychological distress or clinical major depression. The search term *psychometrics* included rating scales, measures, questionnaires, indices, and other related terms and concepts. The final search was confirmed with a professional from the McGill University library.

### Study selection

At least two researchers participated independently in each stage of article screening and full-text selection from which we calculated agreement coefficient (kappa = 0.71, substantial; McHugh, 2012). We included studies that adapted standard validated psychometric measures for depression. We searched for articles that were published in English, as this ensured that we included uniform descriptions of adaptation processes in order to synthesize findings. Articles were included if they (1) were a primary source study, (2) included depression measures, (3) described and/or mentioned the cultural adaptation or validation process of an adapted scale, and (4) had an Indigenous population as its target population. Articles were excluded if (1) they did not include information about the cultural adaptation of a scale, (2) they did not specify the Indigenous identity of the population, (3) they presented adapted scales for non-Indigenous populations, (4) adaptations were done to a scale focused on symptomology unrelated to depression, or (5) they were conference abstracts or poster submissions. The only exception is a protocol for an included adaptation study as it provided supportive information (e.g., cultural adaptation methods) and further context for the full study. The final inclusion and exclusion of full-text articles was confirmed by the first author (LGC).

### Data collection process and data items

Authors systematically and blindly searched for and screened articles (MY, QS, MK, GV, VN, OL). The inclusion of articles followed the Preferred Reporting Items for Systematic Reviews and Meta-analyses extension for Scoping Reviews (PRISMA-ScR) guidelines (Tricco et al., 2018) (Figure 1). Data from the selected articles were extracted and summarized by authors (MY, QS, MK, GV, VN) in two tables. Table 1 includes author(s), publication year, study objectives, setting and location of the research, study methods, language in which the scale was presented, population, sample size, and ethnic identity. Table 2 includes methods of cultural adaptation and the main modifications made to the scales. Following a comprehensive data extraction, we conducted a qualitative and narrative synthesis of the themes by collating, summarizing, and reporting on prominent ideas and concepts from the literature data (Arksey and O'Malley, 2005).

## Results

### Selection of sources of evidence

We identified 3,845 unique articles from the search (following de-duplication) and screened for their titles/abstracts in the first phase. Then, we identified 183 full texts and assessed for eligibility; of these, 37 articles met our inclusion criteria (Chapleski et al., 1997; Ganguli et al., 1999; Bowen and Muhajarine, 2006; Husain et al., 2006; Esler et al., 2007, 2008; Tiburcio Sainz and Natera Rey, 2007; Bass et al., 2008; Campbell et al., 2008; Kaaya et al., 2008; Australian Institute of Health and Welfare, 2009; Fernandes et al., 2011; Mitchell and Beals, 2011; Ekeroma et al., 2012; Brown et al., 2013; Gelaye et al., 2013; Almeida et al., 2014; Armenta et al., 2014; Haroz et al., 2014, 2017; McNamara et al., 2014; Andersen et al., 2015; Sarkar et al., 2015; Schneider et al., 2015; Bougie et al., 2016; Hackett et al., 2016, 2019; Baron et al., 2017; Denckla et al., 2017; Marley et al., 2017; Schantz et al., 2017; Gallis et al., 2018; Harry and Crea, 2018; Kilburn et al., 2018; Ashaba et al., 2019; Chapla et al., 2019; Caneo et al., 2020). Finally, there were 3 cases where the same research team produced two manuscripts detailing adaptation processes; this yielded 34 original reports of cultural adaptation processes included in this review.

### Characteristics of the scale measures with a cultural adaptation (Table 1)

In a total of 34 reports of cultural adaptation processes, there were 41 different depression scales adapted (Table 1). The Indigenous groups of the studies were native to Canada or the United States (6/34), Latin America (3/34), Asia (8/34), Africa (9/34), and Australia or New Zealand (8/34). The most commonly adapted scales included the Patient Health Questionnaire (PHQ-9), the Center for Epidemiologic Studies Depression Scale (CES-D), and the Edinburgh Postnatal Depression Scale (EPDS).

### Methods of scale modifications

#### Revision of cultural relevance and transcultural translation

The cultural relevance of the measurement scales, in addition to the linguistic equivalence (language translation), was verified in 19/34 cultural adaptations (Table 2). An example of such a process was asking Indigenous women about the most relevant tasks and activities related to taking care of themselves, their families, and taking care of and participating in the community. The researchers subsequently translated the mentioned categories and constructed a new instrument with items based off the need to cover sufficient domains of functioning.

Some studies spoke more explicitly about transcultural translation, which was described as a process of understanding the meaning of phrases and ensuring that translation maintains cultural

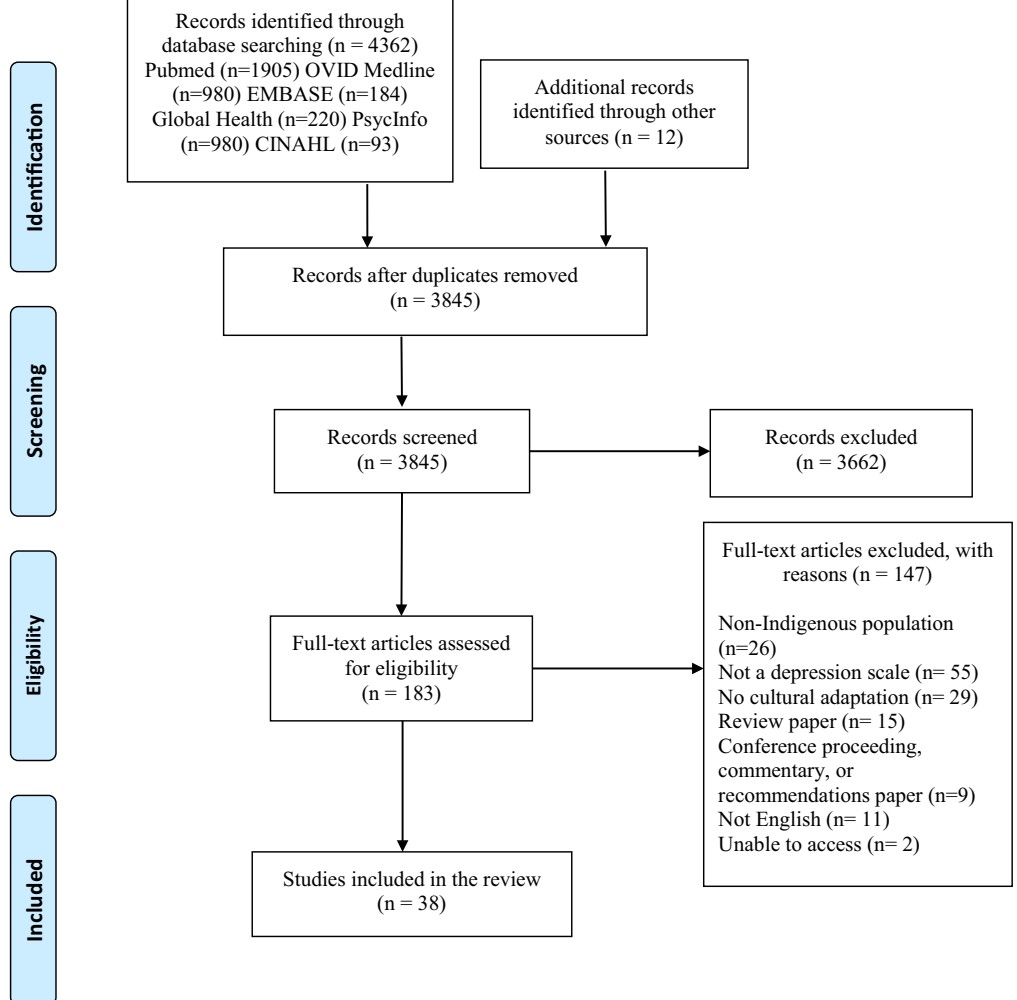

**Figure 1.** Flowchart of search and screening process.

nuances. In all these cases, the researchers used qualitative methods to gain insight into the questions and responses presented in the tools and their relevance from the sociocultural point of view of the target population. *Participatory qualitative methods* were another common characteristic of adaptation processes. This was done in all studies, including through consultations with the Indigenous community focus groups (4/34), qualitative interviews with key community informants (11/34), and or a participatory action research process (1/34) (Table 2). Furthermore, three studies used mixed methods, combining the use of surveys with interviews.

A trend shown by the studies was that significant engagement between researchers/practitioners and Indigenous community members is needed in the process of revising cultural relevance of depression scales. For example, in the cultural adaptation of the Patient Health Questionnaire (PHQ-9) with Indigenous Australian people, focus groups were conducted (Brown et al., 2013). This required the interplay of feedback from bilingual experts and acceptance testing through discussions with community members. The translators, research team members, and Indigenous members discussed the translations to provide linguistic and conceptual equivalences of the adapted scale items, reviewed the difficult items, and identified ways to achieve semantic and cultural equivalence. This process resulted in a modification of words and phrases, as well

as division of one initial question into two items that were more precise (e.g., psychomotor changes and appetite changes). Also, seven key features of depression not covered by the standard PHQ-9 but relevant for the local population were identified and added: anger, homesickness, irritability, excessive worry, weakened spirit, drug/alcohol use, and rumination.

Following the involvement of experts and the community to identify needs, different depression scales were adapted in a variety of ways to increase acceptance of their content and use by the Indigenous group, which improves cultural safety. In studies where items were added, the goal was often to incorporate local idioms of distress or of functioning. For example, the most common somatic symptoms (e.g., 'fatigue') or positive affect concepts (e.g., 'hopefulness') items were added (Baron et al., 2017; Kilburn et al., 2018). Research teams also used a deletion method to remove items perceived as culturally inappropriate or irrelevant to the local constructs of depression. For example, older Australian Aboriginal people perceived the question on the original Kessler Psychological Distress Scale (K-5) concerning feelings of worthlessness as offensive (McNamara et al., 2014). Accordingly, some items about suicidal ideation were subsequently excluded to respect the local sociocultural codes (Gelaye et al., 2013; Schantz et al., 2017). Similarly, the suicidal ideation item was separated into two

**Table 1.** Characteristics of studies

| Measurement | Authors | Study objective(s) | Setting/Context | Location | Language | Study methods | Population | Indigenous group | Sample size |
|---|---|---|---|---|---|---|---|---|---|
| BDI-II[1] HSCL | Ashaba et al. (2019) | Develop, adapt, and validate a depression screening scale | Regional Referral Hospital One HIV Clinic One psychiatric ward One rural community site Rural Areas | Mbarara District, Uganda | Runyankore | Focus groups In-person interviews | Women caregivers/ parents Adults Adolescents (boys and girls aged 13 to 17) Rural population | Banyankore | 25 women caregivers/ parents 15 adolescents (interviews) 35 adolescents (pilot) 224 adolescents (psychometrics) |
| CES-D-20[2] | Armenta et al. (2014) | Examine the longitudinal measurement properties and the utility of the scale Identify the factor structure | Three US reservations Five Canadian First Nations reserves | United States Canada | English | Longitudinal study over 7 years at 8 time points In-person interviews | Adolescents 10 to 12 years | Indigenous North American | 632 |
| CES-D-10[3] | Baron et al. (2017) | Assess the psychometric properties of the scale | Two districts | South Africa | Zulu Xhosa Afrikaans | In-person interviews | Aged 15 years or older | South Africans | 898 |
| CES-D-12[4] CES-D-20 | Chapleski et al. (1997) | Examine the structure of the scale | Urban Rural off-reservation Reservation | Michigan, United States | English | 1st wave of longitudinal study | Elderly 55 years and older | American Indian | 309 |
| CES-D-10[5] | Kilburn et al. (2018) | Assess the psychometric properties of the scale | Rural households | Kenya Malawi Tanzania Zambia Zimbabwe | Bemba /Zambia Chichewa / Malawi Shona /Zimbabwe Swahili /Kenya/ Tanzania | In-home survey In-person interviews | Youth aged 13 to 28 years In extreme poverty | Kenyan Malawian Tanzanian Zambian Zimbabwean | 6,838 (651–2098 per group) |
| CES-D-19[6] | Harry and Crea (2018) | Assess the measurement invariance of the scale | 80 high schools 50 middle schools | United States | English | Longitudinal cohort study In-school surveys In-home interviews | Youth aged 12 to 21 years Adults aged 24 to 34 years | American Indian | 1,154 (654 wave 1; 500 wave 4) |
| CES-D[7] | Andersen et al. (2015) | Enhance knowledge of the subjective experience of depression | Primary infectious disease clinic in a township | Cape Town, South Africa | IsiXhosa English | Semi-structured interviews with the assistance of a translator | Adults living with HIV Between the ages of 25 and 44 Diagnosed with major depression | IsiXhosa speaking | 14 |
| CES-DC[8] | Chapla et al. (2019) | Validate the scale Assess prevalence of depressive symptoms and associated socio-demographic factors | Medium schools | Gujarat, India | Gujarati | In-school survey | Students from 13 to 17 years | Gujarat | 300 (stage 1) 1,000 (stage 2) |

*(Continued)*

**Table 1.** (*Continued*)

| Measurement | Authors | Study objective(s) | Setting/Context | Location | Language | Study methods | Population | Indigenous group | Sample size |
|---|---|---|---|---|---|---|---|---|---|
| CQ[9] SRT CES-D | Tiburcio Sainz and Natera Rey (2007) | Adapt the scales for use in indigenous population through the cognitive laboratories method | One health center of the municipal head town | Municipio del Cardonal,Mexico | Otomi/Ñahñu | In-person interviews | Women aged 16–60 years | Otomi/Ñahñú | 43 (stage 1) 191 (stage 2) |
| DSQ[10] | Kaaya et al. (2008) | Develop a locally specific screen | Primary healthcare antenatal clinics Rural and periurban settings | Chamazi and Mba-gala, Tanzania | English | Qualitative methods Semi-structured in-depth interviews Quantitative analysis | Women of gestational age 28 to 36 weeks Low-income | Tanzanian | 787 |
| EPDS[11] | Bowen and Muhajarine (2006) | Assess the appropriateness and utility of the scale | Centers enrolled in an outreach program | Saskatoon, Canada | English | Interviews | Adult pregnant women Inner-city women | Indigenous Canadian | 39 |
| EPDSb[12] | Campbell et al. (2008) | Describe the development of translation and report psychometric properties | Community services | Townsville, Yapat-jarra and Mt. Isa, Australia | English | Prospective design In-person interviews | Pregnant women and mothers | Indigenous Australian Torres Strait Islanders | 210 |
| EPDS[13] | Ekeroma et al. (2012) | Validate the scale as a screening tool for post-natal depression | One hospital | Auckland, New Zealand | English Tongan Samoan | Self-reported questionnaire In-person interviews | Pregnant women and mothers | Samoan Tongan | 170 |
| EPDS[14] K-10 | Fernandes et al. (2011) | Assess the validity of the scales | Rural prenatal clinic | Karnataka,South India | Kannada | Self-reported question-naires In-person interviews | Pregnant women Rural population | Kannada speaking | 194 |
| EPDS[15] HSCL-D | Bass et al. (2008) | Adapt and validate stand-ard screening instru-ments | One maternity clinic | Kinshasa, Congo | Lingala | In-person interviews | Women Peri-urban popula-tion | Lingala speaking | 80 (stage 1) 133 (stage 2) |
| FAI[16] WHODAS | Schneider et al. (2015) | Developed a locally rele-vant functioning assess-ment instrument to complement a validated instrument Validate the scale | One Community Health Centre | Khayelitsha, Cape Town, South Africa | Xhosa | In-person interviews | Pregnant women Mothers with a baby (under 12 months) Adults (18 years or older) | Xhosa speaking | 40 (stage 1) 142 (stage 2) |
| GDS-H[17] | Ganguli et al. (1999) | Measure depressive symp-tomatology and exam-ine its distribution and associations with age, gender, literacy, cogni-tive and functional impairment | 28 villages Rural communities | Ballabgarh, North-ern India | Haryanvidialect of Hindi | In-person interviews | Illiterate Adults aged 55+ | Hindi-speaking | 1,554 |

(*Continued*)

**Table 1.** (*Continued*)

| Measurement | Authors | Study objective(s) | Setting/Context | Location | Language | Study methods | Population | Indigenous group | Sample size |
|---|---|---|---|---|---|---|---|---|---|
| GDS-15[18] | Sarkar et al. (2015) | Validation of the scale | One village | Puducherry, South India | Tamil | House-to-house survey | Adults aged 60+ Rural population | Tamil speaking | 10 (pilot) 242 (validation) |
| HSCL-25[19] | Haroz et al. (2014) | Development of the scale | Local non-government organizations Community-based organizations | Mae Sot, Thailand | Burmese | In-person interviews | Adult migrants Displaced adults | Burmese | 18 (pilot) 158 (survey) |
| IDSS-G[20] | Haroz et al. (2017) | Develop and assess the reliability and validity of the scale | Two primary health clinics Urban setting | Yangon, Myanmar | Burmese | Qualitative methods Quantitative analysis | Adult patients | Burmese | 147 |
| K-10[21] | Bougie et al. (2016) | Identify the factor structure Assess the reliability of the scale | Living off reserves | Canada | Inuktitut | Aboriginal peoples survey | Youth 15 years or older Adults | First nations Métis Inuit | 17,089 |
| K-6[22] | Mitchell and Beals (2011) | Examine the applicability, psychometric appropriateness and utility of the scale Assess the incremental validity of the scale over and above diagnoses | Two Northern Plains tribes One South-Western tribe | United States | English | Large-scale epidemiological study | Women and men Aged 15–54 years old Living on or within 20 miles of reservation | Two tribal groups American Indians | 3,084 |
| K-5[23] | Australian Institute of Health and Welfare (2009); McNamara et al. (2014) | Assess the psychometric properties of the scale Compare with non-Indigenous individuals | Communities in remote areas | New South Wales state, Australia Torres strait | English | Self-reported questionnaires | Women and men Aged 45 years and older | Australian Indigenous Torres Strait Islanders | 1,939 |
| KICA-dep[24] | Almeida et al. (2014) | Develop a culturally acceptable and valid scale to assess depressive symptoms, associated socio demographic, lifestyle, and clinical factors | Six communities | Western and Derby, Australia | English | Cross-sectional study In-person interviews Comprehensive clinical assessment Semi-structured interviews | Adults aged 45 years or over | Australian Indigenous | 250 |
| KMMS[25] | Marley et al. (2017) | Determine if the scale is reliable, valid, and acceptable Compare the scale to a diagnosis from a blinded clinical expert | 15 sites | Kimberley, Australia | Kimberley English | Cross-sectional Qualitative approach In-person interviews | Pregnant or recently postpartum women Aged 16 years and older | Australian Indigenous | 97 |

(*Continued*)

**Table 1.** (*Continued*)

| Measurement | Authors | Study objective(s) | Setting/Context | Location | Language | Study methods | Population | Indigenous group | Sample size |
|---|---|---|---|---|---|---|---|---|---|
| NOK[26] | Denckla et al. (2017) | Evaluate the reliability, validity, and factor structure of the instrument | Rural schools | Makindu and Machakos, Kenya | Kiswahili Kikamba | Self-reported questionnaires | Children aged 10 to 18 years old | Kiswahili speaking Kikamba speaking | 2,282 |
| PHQ-9[27] | Brown et al. (2013) | Translate and adapt the scale into different dialects | Remote communities | Alice Springs and Central Australia, Australia | Pitjantjatjara Luritja Arrernte Anmatyerre Warlpiri | Qualitative methods Focus groups In-depth interviews | Adults | Arrernte- Pitjantjarjara Anmatyerr Warlpiri Pintupi /Luritja | 22 |
| PHQ-9[28] | Hackett et al. (2016); Hackett et al. (2019) | Determine the validity, sensitivity, specificity, and acceptability of the scale | 10 primary healthcare services Urban areas Rural and remote areas | Australia | English | Questionnaire administered by primary health provider | Adult patients | Australian Indigenous Torres Strait Islander | 500 |
| PHQ-10[29] | Esler et al. (2007); Esler et al. (2008) | Assess the acceptability, reliability, and validity of the scale Determine the prevalence of depression in the population | One community health service | Darwin, Australia | English | Qualitative methods Focus-groups Semi-structured clinical interviews | Adult patients with Ischemic Heart Disease Trainee health workers | Australian Indigenous Torres Strait Islander | 67 (34 patients, 33 health workers) |
| PHQ-8[30] | Schantz et al. (2017) | Assess the validity and reliability of the scale | Three public hospitals | La Paz and El Alto, Bolivia | Spanish | In-person interviews | Adult outpatients with hypertension and/or diabetes Low- income | Andean Indigenous | 107 |
| PHQ-9[31] | Gallis et al. (2018) | Demonstrate the criterion-related validity and internal reliability of the scale | Rural communities | Pakistan | Urdu | In-person interviews | Adults Pregnant women Rural women | Urdu speaking Punjabi speaking Potohari speaking | 1731 |
| PHQ-9[32] | Gelaye et al. (2013) | Evaluate the reliability and validity of the questionnaire | Outpatient departments of a major referral hospital | Addis Ababa, Ethiopia | Amharic | In-person interviews | Adults (≥ 18 years of age) | Amharic speaking | 926 (stage 1) 384 (stage 2) |
| PHQ-9[33] SRQ | Husain et al. (2006) | Select the best threshold scores for the scales Compare the sensitivity and specificity of the scales Examine the influence of socio-demographic factors on misclassification | Rural communities | Mandra, Pakistan | Urdu | House-to-house surveys In-person interviews | Rural population | Potohari or Urdu speaking | 258 (stage 1) 107 (stage 2) |

(*Continued*)

**Table 1.** (*Continued*)

| Measurement | Authors | Study objective(s) | Setting/Context | Location | Language | Study methods | Population | Indigenous group | Sample size |
|---|---|---|---|---|---|---|---|---|---|
| PHQ-2[34] | Caneo et al. (2020) | Assess the performance and validity of the scale | Rural communities | Maule, Chile | Spanish | In-person interviews | Adults aged 38–74 years Agricultural population Rural population | Chilean | 4,767 |

[1]Beck Depression Inventory-II (BDI-II), and Hopkins Symptom Check List (HSCL).
[2]Center for Epidemiologic Studies Depression Scale – 20 items (CES-D-20).
[3]Center for Epidemiologic Studies Depression Scale – 10 items (CES-D-10).
[4]Center for Epidemiologic Studies Depression Scale– 12 items (CES-D-12) and Center for Epidemiologic Studies Depression Scale– 20 items (CES-D-20).
[5]Center for Epidemiologic Studies Depression Scale – 10 items (CES-D-10).
[6]Center for Epidemiologic Studies Depression Scale – 19 items (CES-D-19).
[7]Center for Epidemiologic Studies Depression Scale (CES-D).
[8]Center for Epidemiologic Studies Depression Scale for Children (CES-DC).
[9]Coping Questionnaire (CQ), Symptom Rating Test (SRT), and Depression Scale Center for Epidemiological Studies (CES-D).
[10]Dar-es-Salaam Symptom Questionnaire (DSQ).
[11]Edinburgh Postnatal Depression Scale (EPDS).
[12]Edinburgh Postnatal Depression Scale (EPDS).
[13]Edinburgh Postnatal Depression Scale (EPDSc).
[14]Edinburgh Postnatal Depression Scale (EPDS), Kessler Psychological Distress Scale (K-10).
[15]Edinburgh Postnatal Depression Scale (EPDS), Hopkins Symptom Checklist (HSCL-D).
[16]Functioning Assessment Instrument (FAI), World Health Organization's Disability Assessment Schedule (WHODAS).
[17]Hindi version of the Geriatric Depression Scale (GDS-H).
[18]Geriatric Depression Scale – 15 items (GDS-15).
[19]Hopkins Symptom Checklist Depression Scale (HSCL-25).
[20]International Depression Symptom Scale-general (IDSS-G).
[21]Kessler Psychological Distress Scale – 10 items (K-10).
[22]Kessler Psychological Distress Scale – 6 items (K-6).
[23]Kessler Psychological Distress Scale – 5 items (K-5).
[24]Kimberley Indigenous Cognitive Assessment of Depression (KICA-dep).
[25]Kimberley Mum's Mood Scale (KMMS).
[26]Ndetei–Othieno–Kathuku Scale (NOK).
[27]Patient Health Questionnaire – 9 items (PHQ-9).
[28]Patient Health Questionnaire – 16 items (PHQ-16).
[29]Patient Health Questionnaire – 10 items (PHQ-10).
[30]Patient Health Questionnaire – 8 items (PHQ-8).
[31]Patient Health Questionnaire – 9 items (PHQ-9).
[32]Patient Health Questionnaire – 9 items (PHQ-9).
[33]Patient Health Questionnaire – 9 items (PHQ-9).
[34]Patient Health Questionnaire – 2 items (PHQ-2).

**Table 2.** Process of cultural adaptation

| Measurement scale | Authors | Methods | Modifications |
|---|---|---|---|
| BDI-II[1] HSCL | Ashaba et al. (2019) | Focus group discussions<br>One-on-one in-depth interviews<br>Transcultural translation<br>Mixed methods | *Items Added:* 'having many thoughts', 'being desperate', 'loss of hope', 'self-hatred', 'weight loss', 'loneliness', 'anger', 'uselessness', 'lack of confidence', 'forgetfulness', 'lack of peace', 'frustration with life', 'feeling stressed', 'suicidal thoughts'<br>*Items Removed:* changes in sleeping patterns, appetite, and/or sexual interest<br>*Item Modification:*<br>• 14 items were adapted from the HSCL: fearful, fidgety, worthlessness, hopelessness, loneliness, self-blame, loss of interest in things, low energy and not caring about one's own health<br>• 14 items were adapted from the BDI-II: feeling like a failure, loss of pleasure, guilt, self-dislike, self-blame, crying easily, pessimism, feelings of being punished, being self-critical, agitation, change in sleeping patterns, changes in weight, and loss of interest in sex |
| CES-D-20[2] | Armenta et al. (2014) | Revision of linguistic equivalence and cultural relevance<br>Qualitative interviews | *Conceptual*: recommendation to review notion of loneliness<br>*Items Added:* recommendation to add somatic difficulties items<br>*Items Removed:* potential inadequacy of the question 'I felt everything I did was an effort' |
| CES-D-10[3] | Baron et al. (2017) | Standard translation<br>Qualitative interviews | *Conceptual*: caution about conceptualizing question about hopefulness as a positive affect concept<br>*Organizational*: reordering the questions, from positive to negative |
| CES-D-12[4] CES-D-20 | Chapleski et al. (1997) | Surveys<br>Focus groups discussions<br>Key-informant interviews<br>Revision of cultural relevance | *Items Removed:* concerning interpersonal factors |
| CES-D-10[5] | Kilburn et al. (2018) | Standard translation<br>Qualitative interviews | *Item Removed:* recommendation to review the question 'everything was an effort'<br>*Items Added:* Recommendation to include an additional somatic element<br>*Conceptual:* 'Hopeful' may not perfectly align with the conceptualization of positive affect |
| CES-D-19[6] | Harry and Crea (2018) | Revision of linguistic equivalence and cultural relevance<br>Survey<br>Interviews | *Items Added:* 'you felt that life was not worth living'<br>*Semantic:* pronoun 'I' was changed to 'you'<br>*Response categories change:* 'never or rarely', 'sometimes', 'a lot of the time', 'most of the time/all of the time'<br>*Item Modification:*<br>• 'you felt that you were too tired to do things' to 'it was hard to get started doing things'<br>• inclusion of 'friends' to 'I felt that I could not shake off the blues even with help from my family' |
| CES-D[7] | Andersen et al. (2015) | Qualitative interviews with the assistance of a translator | *Conceptual:*<br>• Identification of several idioms of distress that could assist in screening for depression.<br>• Although the construct of depression was consistent with DSM-IV criteria, the symptom presentation was distinctive |
| CES-DC[8] | Chapla et al. (2019) | Standard translation | *Semantic/Conceptual:* evaluation of the semantic and content equivalence of the scale (details were not reported) |
| CQ[9] SRT CES-D | Tiburcio Sainz and Natera Rey (2007) | Revision of linguistic equivalence and cultural relevance | *Semantic:*<br>• for CQ: phrasing of 26 items changed except for #3, 4, 9 and 26<br>• for SRT: nearly all items modified (no further details)<br>• for CES-D: 9 items were modified (no further details) |
| DSQ[10] | Kaaya et al. (2008) | Revision of linguistic equivalence and cultural relevance<br>Qualitative interviews | *Items added:* addition of 30 local idioms for depression and anxiety |
| EPDS[11] | Bowen and Muhajarine (2006) | Development of a protocol for administration:<br>during a regular home visit<br>availability of the nurse to clarify questions and discuss the score with the patient<br>Qualitative interviews | *Items Added:*<br>• questions about alcohol, tobacco, and drug use<br>• addition of two questions about social support |
| EPDS[12] | Campbell et al. (2008) | Participatory action research<br>Focus group discussions | *Semantic changes:* alteration of some words and simplification of sentence structures |

*(Continued)*

**Table 2.** (*Continued*)

| Measurement scale | Authors | Methods | Modifications |
|---|---|---|---|
| | | Consultations with community members, service staff, and board | *Item Modification:*<br>• 'I stress out for no good reason or I feel like going wombat/wongie for no good reason', 'I get sick in the guts for no good reason or I get scared and sick and don't know why', 'I feel frightened and shaky all the time', 'Sometimes I feel like doing something stupid or I think about killing myself', 'I feel really no good', 'I feel things are getting me down, I need a rest' |
| EPDS[13] | Ekeroma et al. (2012) | Standard translation<br>Interpreters | • Appropriateness of language and meaning were checked |
| EPDS[14]<br>K-10 | Fernandes et al. (2011) | Standard translation | • Conceptual equivalent of phrases was checked |
| EPDS[15]<br>HSCL-D | Bass et al. (2008) | Qualitative interviews<br>Mixed methods<br>Discussions with Key Informants<br>Transcultural translation | *Item Removed:*<br>• 'you have been able to laugh when something is funny', 'you have looked forward to the future with enjoyment'<br>*Items Added:*<br>• 12 activities as a measure of functional impairment: Commerce, Helping others, Working gardens, Visiting the sick, Preparing meals, Taking care of the home, Washing clothes, Taking care of one's body, Taking care of one's hair / braids, Bathing in hot water, Washing the diapers, Washing the new baby<br>• Sign and symptoms: Restless /agitated heart, Low in energy / fatigued / tired, Angry, Lack of peace, Tormented, Self-pity, Stomach pains, Disputing / arguing for no reason |
| FAI[16]<br>WHODAS | Schneider et al. (2015) | Qualitative interviews<br>Revision of linguistic equivalence and cultural relevance<br>Consultations with community members | *Items Added:*<br>• 9 items: Cleaning the house, Preparing and cooking food for the family, Doing laundry, Bathing yourself, Bathing babies and children, Taking care of the needs of babies and children (feeding, preparing for crèche or school, taking children to crèche and school, keeping them safe, etc.), Taking care of emotional needs of babies and children (loving them, playing with them, helping with homework, etc.), Spending time and doing activities with family and friends, Exercising |
| GDS-H[17] | Ganguli et al. (1999) | Standard translation | *Item Modification:* the item asking whether the individual found life to be exciting was altered to substitute 'very enjoyable' |
| GDS-15[18] | Sarkar et al. (2015) | Standard translation | Not reported |
| HSCL-25[19] | Haroz et al. (2014) | Revision of linguistic equivalence and cultural relevance<br>Qualitative interviews | *Items Added:*<br>• 'hopeless', 'do not care what will happen, 'disappointment', 'heart beating quickly' |
| IDSS-G[20] | Haroz et al. (2017) | Revision of linguistic equivalence and cultural relevance<br>Qualitative interviews | *Items Added:* addition of 16 questions on symptoms such as: crying, lonely, social withdrawal, worry, worthless, headaches, stomachaches, general aches and pain, anger, thinking too much, confused, heart weakness, palpitations, heavy heart, heart pain<br>*Semantic:* participants found items 'feeling weakness in your heart' and 'feeling as though your heart was heavy' difficult to understand |
| K-10[21] | Bougie et al. (2016) | Standard translation | Not reported |
| K-6[22] | Mitchell and Beals (2011) | Revision of linguistic equivalence and cultural relevance<br>Focus group discussions | Not reported |
| K-5[23] | Australian Institute of Health and Welfare (2009); McNamara et al. (2014) | Revision of linguistic equivalence and cultural relevance<br>Qualitative interviews | *Item Removed:* 'feel worthless' was found to be offensive and omitted<br>*Semantic:* word changes to items to improve their understanding:<br>• 'restless or jumpy', 'without hope', 'last four weeks' |
| KICA-dep[24] | Almeida et al. (2014) | Revision of linguistic equivalence and cultural relevance<br>Clinical assessment<br>Qualitative interviews | *Item modification:*<br>• question about suicide ideation presented as two separate questions: 'have you had thoughts that you would be better off dead?' and 'have you thought of hurting yourself?' |
| KMMS[25] | Marley et al. (2017) | Revision of linguistic equivalence and cultural relevance<br>Use of a visual scale<br>Editing a Protocol for administration<br>Development of a culturally safe space<br>Qualitative interviews<br>Focus groups | *Semantic:* rewording of the questions (no further details) |

**Table 2.** (*Continued*)

| Measurement scale | Authors | Methods | Modifications |
|---|---|---|---|
| NOK[26] | Denckla et al. (2017) | Standard Translation<br>Compiling a list of symptoms commonly encountered in the clinical work | *Items Added:* 33-items: 'How much were you distressed by': 'feeling as if insects or ants are crawling under your skin', 'feeling your heart has fallen down', 'feeling a pressure on the top of your head', 'getting frequent attacks of malaria', 'often having pain in your bones', 'feeling your heart is heavy'. |
| PHQ-9[27] | Brown et al. (2013) | Revision of linguistic equivalence and cultural relevance<br>Qualitative interviews | *Semantic:* questions were rephrased<br>*Item Modification:* certain items were confusing and separated into two elements: 'psychomotor', 'appetite changes' |
| PHQ-9[28] | Hackett et al. (2016); Hackett et al. (2019) | Revision of linguistic equivalence and cultural relevance<br>Qualitative interviews | *Items Added:* inclusion of seven additional questions assessing: 'anger', 'weakened spirit', 'homesickness', 'irritability', 'excessive worry', 'rumination', 'drug/alcohol use' |
| PHQ-10[29] | Esler et al. (2007); Esler et al. (2008) | Revision of linguistic equivalence and cultural relevance<br>Improvement of the Protocol for administration<br>Development of a culturally safe space<br>Clinical assessment<br>Qualitative interviews | *Semantic:* modification of the wording of some questions<br>*Items Added:* inclusion of one question pertaining to anger (a culturally specific symptom of depression) |
| PHQ-8[30] | Schantz et al. (2017) | Standard translation<br>Interviews | *Items removed:* exclusion of one question about suicide<br>*Pragmatic:* caution about question 'difficulty concentrating in reading' because it is not sensitive to low literacy |
| PHQ-9[31] | Gallis et al. (2018) | Standard translation<br>Revision of linguistic equivalence and cultural relevance | *Semantic:* some difficult Urdu words were replaced by simpler and more frequently used words carrying the same meaning |
| PHQ-9[32] | Gelaye et al. (2013) | Standard translation | *Semantic:* ensure proper expression and conceptualization of terminologies in local contexts<br>*Items added*: one additional item (#10) assessing functional impairment |
| PHQ-9[33]<br>SRQ | Husain et al. (2006) | Standard translation | *Items removed:* Psychosexual items in the interview had to be omitted in most female interviews |
| PHQ-2[34] | Caneo et al. (2020) | Standard translation | Not reported |

[1]Beck Depression Inventory-II (BDI-II), and Hopkins Symptom Check List (HSCL).
[2]Center for Epidemiologic Studies Depression Scale– 20 items (CES-D-20).
[3]Center for Epidemiologic Studies Depression Scale – 10 items (CES-D-10).
[4]Center for Epidemiologic Studies Depression Scale– 12 items (CES-D-12) and Center for Epidemiologic Studies Depression Scale– 20 items (CES-D-20).
[5]Center for Epidemiologic Studies Depression Scale – 10 items (CES-D-10).
[6]Center for Epidemiologic Studies Depression Scale – 19 items (CES-D-19).
[7]Center for Epidemiologic Studies Depression Scale (CES-D).
[8]Center for Epidemiologic Studies Depression Scale for Children (CES-DC).
[9]Coping Questionnaire (CQ), Symptom Rating Test (SRT), and Depression Scale Center for Epidemiological Studies (CES-D).
[10]Dar-es-Salaam Symptom Questionnaire (DSQ).
[11]Edinburgh Postnatal Depression Scale (EPDS).
[12]Edinburgh Postnatal Depression Scale (EPDS).
[13]Edinburgh Postnatal Depression Scale (EPDS).
[14]Edinburgh Postnatal Depression Scale (EPDS), Kessler Psychological Distress Scale (K-10).
[15]Edinburgh Postnatal Depression Scale (EPDS), Hopkins Symptom Checklist (HSCL-D).
[16]Functioning Assessment Instrument (FAI), World Health Organization's Disability Assessment Schedule (WHODAS).
[17]Hindi version of the Geriatric Depression Scale (GDS-H).
[18]Geriatric Depression Scale – 15 items (GDS-15).
[19]Hopkins Symptom Checklist Depression Scale (HSCL-25).
[20]International Depression Symptom Scale- General (IDSS-G).
[21]Kessler Psychological Distress Scale – 10 items (K-10).
[22]Kessler Psychological Distress Scale – 6 items (K-6).
[23]Kessler Psychological Distress Scale – 5 items (K-5).
[24]Kimberley Indigenous Cognitive Assessment of Depression (KICA-dep).
[25]Kimberley Mum's Mood Scale (KMMS).
[26]Ndetei–Othieno–Kathuku Scale (NOK).
[27]Patient Health Questionnaire – 9 items (PHQ-9).
[28]Patient Health Questionnaire – 16 items (PHQ-16).
[29]Patient Health Questionnaire – 10 items (PHQ-10).
[30]Patient Health Questionnaire – 8 items (PHQ-8).
[31]Patient Health Questionnaire – 9 items (PHQ-9).
[32]Patient Health Questionnaire – 9 items (PHQ-9).
[33]Patient Health Questionnaire – 9 items (PHQ-9).
[34]Patient Health Questionnaire – 2 items (PHQ-2).

questions in another study: *Have you had thoughts that you would be better off dead?* and *Have you thought of hurting yourself?* (Almeida et al., 2014). Thus, the desire of the community was most important when considering including questions about suicide in depression measures.

Other than through sole use of qualitative methods, some studies adapted scales using quantitative methods. Following interviews with mothers, Bass et al. (2008) compiled a list of identified postpartum symptoms and statistically regressed them on mental health constructs that comprised the measure (Bass et al., 2008). Through this, the authors identified relevant depression-like problems that were meaningful based on both phenomenological insight and objective validation. Subsequently, they reviewed standard post-partum depression screens to identify those that best reflected the local descriptions. Based on this work, they selected, translated, and adapted the EPDS and the Hopkins Symptom Checklist depression section (HSCL-D). This involved adding new questions associated with signs and symptoms that were not part of the original screens but were described by the interviewees.

### Language translation

Translation to the respective Indigenous language was completed in 14/34 scales (Table 2). In general, these translations followed a rigorous protocol, informed by guidelines and examples set forth by experts in psychiatry or experts in the field of linguistics (Ganguli et al., 1999; Brown et al., 2013; Almeida et al., 2014). Most translations were done by professional translators with recognized experience in psychiatry; in two cases, non-native translators were fluent in both the original scale language and the Indigenous language (Easton et al., 2017; Schantz et al., 2017). Additionally, the first translated version of certain tools was completed in a blinded manner and through back-translation (14/14). When discrepancies arose, the researchers turned to mental health experts to define the most accurate translation (9/14).

Following a direct transcultural translation, some scale items needed to be rewritten for grammatic purposes. This necessitated reformulation of items (2/14), addition of items (2/14), removal of items (2/14), changes to the order of questions (1/14), and recommendations regarding the wording of items (5/14). To assist in this, some studies compiled a list of Indigenous symptoms or expressions of distress that could be useful for screening for depression. This information was collected through qualitative methods including interviews (4/14) and use of interpreters (1/14). An example of a language translation process is reported by Ganguli et al. (1999). They translated the Geriatric Depression Scale (GDS), originally written in English, into the local Haryanvi dialect of Hindi through a standard approach of iterative back-translation by bilingual clinicians (Ganguli et al., 1999). The researchers also made minimal modifications to the content of the tool to improve cultural appropriateness. The question regarding whether the person found life to be 'exciting' was modified to 'very enjoyable', as the concept of older people feeling 'excited' was found to not be appropriate in rural India. The adapted scale was named the Geriatric Depression Scale-Hindi version (GDS-H).

### Visual scales

One study developed a visual scale to accompany the original numerical Likert scale (Marley et al., 2017). In this study, the Kimberley Mum's Mood Scale (KMMS) was adapted to fit the needs of pregnant and postpartum Indigenous women in Australia. The qualitative methodology relied on multiple focus groups for input on appropriate visual representations, wording, and a protocol for administration of the tool. The resulting visual scale included a water glass pictorial to distinguish different multiple-choice responses, pictographs, and a visual analogue scale comprised of degrees of facial expression indicating valence and magnitude of emotions.

### Scale administration

Increasing cultural safety and sensitivity at administration of the scale was often the aim of the studies' adaptation processes. The context of use as well as the scale administration process were considered by three studies (Table 2). Some authors tried to improve the acceptability, reliability, and utility of the scales, and explored impact of the administration process on this. Increasing *acceptability* of the tool by the Indigenous community was described as being the process whereby adapted tools were considered sufficiently culturally safe, non-judgmental, and could provide providers with valuable insights into participants' lived experiences and social, emotional, and cultural well-being. Increasing acceptability also supports the *validity* of the scale, which is the accuracy of the tool for measuring constructs of depression that align with the Indigenous' conceptualizations of it.

Esler et al. (2007) investigated attitudes toward depression screening and the components of the PHQ-9 among members of an urban Indigenous community in Australia (Esler et al., 2007). Some focus group participants suggested that involving a family member in the assessment process may improve the accuracy of the screening while other participants shared the opposite point of view. Most participants stated they would prefer an Indigenous health worker as interviewer. Equally, familiarity with the Indigenous community-controlled health service was considered to contribute to the acceptance of the administration of the tool. The need for an initial trusting relationship between the patient and the tool administrator was noted as being particularly important.

In Marley et al.'s (2017) adaptation of the KMMS, the study team created a culturally safe space to conduct the screen with the Indigenous women (Marley et al., 2017). They incorporated a guided questionnaire that allowed the women participants to describe their history in detail, making it easier for researchers and clinicians to identify protective and vulnerability factors. The authors also developed culturally safe training modules for staff administering the tool. Resultingly, the majority of participants reported that completing the KMMS was highly useful and better than the standard instrument (EPDS), as it increased mutual respect, trust, and understanding between the personnel and the participants.

## Discussion

We identified 34 reports on cultural adaptation processes including 41 different depression scales. The most commonly adapted scales included the Patient Health Questionnaire (PHQ-9), the Center for Epidemiologic Studies Depression Scale (CES-D), and the EPDS. The review found that cultural adaptation of depression scales is more of an exception than a rule; only 34 reported cultural adaptations were identified, yet the estimated global number of Indigenous groups is over 5,000 (United Nations, 2023). Geographically, the cultural adaptations were mostly conducted in Africa, Australia, and Asia. This finding is consistent with the geographic dispersion of Indigenous populations, as approximately 70% of Indigenous populations live in Asia and the Pacific, 16.3% in

Africa, 11.5% in Latin America and the Caribbean, 1.6% in North America, and 0.1% in Europe and Central Asia (Jamil Asilia Centre, 2021). This shows a heightened need for attention to recognizing importance of cultural safety in mental health care for Indigenous populations of these areas.

Four main processes of utilized cultural adaptation of depression scales were identified. The most commonly used non-exclusive methods were revision of cultural appropriateness or transcultural psychiatry, language translation, modifications of the administration process, and complementary visual scales. Most importantly, the approach chosen depended on the context of administration and the specific clinical or research issues to address. Cultural adaptations to scales could increase cultural safety and response rate of psychometrics; however, it remains open when it is appropriate to consider conducting a translation or further cultural adaptation. A careful, shared needs assessment is important before cultural adaptations, given the intensive use of Indigenous resources in the adaptation process.

### Importance of cross-cultural translation

Studies found that a rigorous translation of the tool into the native language was important, since the language is a significant identity marker for the community members. As compared to literal equivalence in traditional translation, the main challenges in cultural translation are the complexity of local semantics and the need to guarantee conceptual equivalence, criterion equivalence, and content equivalence (Ghimire et al., 2013). The success of translation processes necessitates several steps and good resources: (1) establishment of a bilingual group of experts, (2) examination of conceptual composition by the experts, (3) translation by an expert, (4) examination of the translation by the experts, (5) examination of the translation by a group representative of the population, (6) blind back-translation by another expert, (7) examination of the blind back-translation by the experts and adjustment of discrepancies, (8) a pilot study to test the tool among the population, and (9) adjustments based on pilot study results (Sartorius and Janca, 1996; Beaton et al., 2000; Sousa and Rojjanasrirat, 2011; Arafat et al., 2016). This process can last up to several months and constitutes the most simple and economical form of adaptation processes.

### Importance of qualitative methods

Confirming that a cultural adaptation process is successful and useful to the population is important to support its use (Wiltsey Stirman et al., 2019). In quantitative data collection, completion rates for individual items can provide indirect evidence for the acceptance of a tool. While quantitative research provides a rapid insight into scale-user satisfaction, these methods do not provide concrete information for improvement of cultural sensitivity (Ingersoll-Dayton, 2011). For instance, if items are deemed unclear, researchers must understand culturally relevant ways to modify them. To do this, qualitative methods are necessary, as is following-up on user experiences (Ingersoll-Dayton, 2011). Iterative focus groups, fieldwork, qualitative interviews, and feedback collection with community members and clinicians (among other methods) are appropriate methods to review and improve different iterations of a scale until a final, culturally accepted tool is disseminated for community use (Ingersoll-Dayton, 2011; Brown et al., 2013).

### Response options

Colonization, repeated systemic discrimination, and unsafe schools have limited access to resources and conditions necessary to maximize socioeconomic status for many Indigenous groups (Reading and Wien, 2009; Davy et al., 2016; Gall et al., 2021). This disadvantage can be manifested in lower rates of educational attainment, literacy, and familiarity of colonizers' language, which limit reliability of responses to scales and influence the accuracy of depression screening instruments (Akena et al., 2012). As an example, high rates of missing data on Likert response options have challenged defining the diagnosis (Bell et al., 2005). Conversely, participants' openness to respond can be compromised when the native language is not used (Borsa et al., 2012; Ghimire et al., 2013; Arafat et al., 2016). Thus, translation, audio recordings of the questions, or administration of the tool verbally can improve reliability.

### Indigenous co-leadership in scale development

Seeking Indigenous knowledge to inform decision-making on selection, cultural adaptations, and administration of scales, implies that Indigenous peoples are stakeholders in their own mental health (Latulippe and Klenk, 2020). Reconciliation is about the genuine restructuring and transformation of the relationships between Indigenous and settler people (Hoicka et al., 2021). Reconciliation can take place in cultural adaptation processes that use Indigenous knowledge as 'expert' knowledge. In fact, through qualitative interviews to learn about experiences with culturally adapted tools, Indigenous people reported that adapted tools 'mean something to them', and that these scales can make Indigenous 'identify with their own person within' (Marley et al., 2017).

Prior to cultural adaptation, Indigenous peoples in the studies reported that the original scales created "distrust and questions before you get started, and that confidence is really hard to get back" (Marley et al., 2017). In contrast, inviting Indigenous people in the leadership of their own tools supported trust and mutual respect of researchers/practitioners and Indigenous groups, which allows for confidence in and readiness for cultural adaptations. In this way, the studies illustrated how Indigenous co-leadership in scale development was a determinant of the success of other adaptation methods. This is because this process facilitated collaboration in translation and revision of cultural relevance, as well as established a trusting relationship when Indigenous peoples were invited to focus group discussions, key informant interviews, or consultations.

Indigenous co-leadership in adapting scales requires time and resources from both researchers and the Indigenous community. This is stressing the importance of shared needs assessment before the process of cultural adaptation. The needs assessment has to start from the overall idea of using symptom measures. For symptom measures to be useful, cultural safety cannot compromise utility for the medical approach in evaluation and treatment, necessary to inform mental health services.

### Holistic constructs of mental well-being

Some tools were developed to be complementary to the symptom measure scales, considering Indigenous conceptualizations of well-being and honoring the self-determination of the Indigenous group. For example, this was the case of the Functional Assessment Instrument (FAI), developed through *free-listing interviews* – a particular technique that explores how a group thinks about and

defines health domains through engaging communities to identify their shared priorities from the perspective of their own ideologies (Schneider et al., 2015). The researchers adapted this scale to identify tasks and activities that women performed during the perinatal period, which they expressed was more relevant to them than clinical features of post-partum assessments and were more meaningful for healing.

Several Indigenous groups reported feeling that alternatives to symptom measures were more consistent with Indigenous visions of health. This aligns with the literature sharing that Indigenous ideologies embrace a holistic concept of health that reflects physical, spiritual, emotional, and mental dimensions (Reading and Wien, 2009). Accordingly, other than symptom scales adapted by studies included in the review, there exists tools to evaluate distress while considering factors specific to the sociocultural reality of the community and their worldviews (Kinzie et al., 1982). For example, the Native American Cultural Values and Beliefs Scale assesses whether the individual experiences any distress based on their lack of participation in important Dakota, Nakota, and Lakota values and beliefs (Reynolds et al., 2006). Other instruments have moved toward the assessment of more general factors that promote or hinder mental health in a broader sense (e.g., emotional wellness and empowerment) rather than standard clinical measures (Haswell et al., 2010). Moreover, some scales for Indigenous groups have moved away from assessing negative constructs of mental illness, and toward strengths and resources associated with resilience, healing, and recovery (Gee et al., 2023).

Inclusion of local cultural expressions can be particularly important in depression, which is a psychological and experiential phenomenon whose meanings are closely shaped by the sociocultural context. Moreover, measurement scales for Western populations may contain behaviors or experiences considered normal or irrelevant to the local culture of the Indigenous population.

Western symptom measures have been developed based on an individual view of health; cultural adaptation of measures cannot change this fact if the focus is still on symptomology. In contrast, if a screen is to be truly aligned with the local Indigenous concept of health, it is possible that it is developed less as a screen of symptomology but rather developed holistically, recognizing that *health* is interrelated with and interconnected to their families, community, and lands.

### Cultural safety during the adaptation and use of the scale

Researchers have emphasized the importance of an ethnographic approach when assessing global mental health and providing care. The incorporation of qualitative and ethnographic methods in developing assessment tools and interventions is well established and recognized (Campbell et al., 2000; Moreau et al., 2009; Onwuegbuzie et al., 2010). Collaborative-participatory research, commitment of Indigenous peoples, and a relationship of trust and respect with the community are crucial methods for a successful process of cultural adaptation. The ethical principles that regulate the research with Indigenous populations must be respected by the research team. Respecting the national standards as well as the specific standards of the target Indigenous group is necessary and essential to confirm further use of adapted depression screens.

Increasing cultural safety starts in the planning stage for conducting research or developing programs for Indigenous peoples, and inclusion of the target population is essential from the early stages of needs assessment and planning (Kirmayer et al., 2003; Darroch et al., 2017; Yaphe et al., 2019). For example, it is recommended to have research members or service providers of the same ethnic origin as the respondents available to establish a safe space (Rait et al., 1999; Kaiser et al., 2013; Billan et al., 2020). When this is not possible, the administrators of the tools may undergo training in cultural competency through workshops, educational programs, or continuing education to create a culturally sensitive space which correspondingly facilitates communication (Billan et al., 2020). Finally, it is essential to consider institutional discrimination and possible stigma of psychiatric diagnoses in the administration of scales. In cultural adaptation processes, minding the local principles are important as it addresses the ownership of data of adapted scale, as well as the patient's definition of what safe care means (Baba, 2013).

### Unanswered questions and future research

While the final version of a culturally adapted depression psychometric remains specific for the target community, scientifically reporting the process of scale cultural adaptation makes it possible to evaluate methods and make conclusions about culturally specific factors that promote or threaten mental health. Reports also enable accumulation of knowledge on suitable methods not only for culturally sensitive measurement but also to inform treatment.

Deleting unsafe items can also have limitations. The lack of inclusion of 'idioms of distress' or 'cultural concepts of distress' in the diagnoses developed by the biomedical approach can result in low levels of reliability in the information communicated by people. This can potentially interfere with appropriate diagnoses and securing appropriate mental health services or support (Nichter, 2010; Kohrt et al., 2016; Kidron and Kirmayer, 2019; Lewis-Fernández and Kirmayer, 2019). Thus, it should be noted that adaptations that seem necessary to increase acceptability might compromise other psychometric characteristics and thus, confirming validity, comparability, and clinical utility remains an equally important but separate line of development, where normal guidelines for psychometric evaluation apply.

Currently, selection of populations for cultural adaptation seems to have been arbitrary, dependent on motivation of researchers and community members. Future work may include a needs assessment before devoting external and community resources to a cultural adaptation. A rigorous process of cultural adaptation is challenging and necessitates time and resources. Recognizing the populations where measurement will not be possible without a translation or a more profound cultural adaptation as well as prioritizing populations for adaptation based on a needs assessment might be necessary. Alternatively, focus on culturally safe administration of standard scales might be an option.

Greater focus on including supporting visuals (e.g., pictographs) and auditory materials would extend the accessibility of these tools to low literacy or small populations. Sometimes, valid scales could be modified to avoid the most culturally contradictory items. Research should optimally compare acceptability and validity of pure translations to cultural adaptation of the same scales in the same population and calculate the resources vs. benefit of the process.

Use of cultural knowledge to complement mental health interventions is an interesting approach that should be explored. The process of cultural adaptation, and knowledge about local concepts

in mental health and the culture specific resilience, can also indirectly increase cultural safety and respect of minorities.

## Strengths and limitations

To our knowledge, this is the first systematic scoping review on methods used for culturally adapting depression measures with Indigenous peoples. Broadly, our analysis contributes to the field of cross-cultural studies and to cultural psychiatry providing well established practices that can guide the process of cultural adaptation of measurement tools, as proposed recently (Arafat et al., 2016). The findings of this review may facilitate further research and more equitable exchanges of knowledge on culturally sensitive measures for not only psychiatry but also resilience and supporting factors. The latter remains a generally unmet need; reports have shown that many psychometric scales can be experienced as unsafe and that empowering approaches to mental health assessment are preferred by several Indigenous groups (Gomez Cardona et al., 2021).

Our search was limited to articles in English. We discarded several articles due to the complexity of the definition of Indigenous peoples and the way this was addressed in the research. In fact, the definition of *Indigenous* is problematic since the auto-definition of individuals as such is not always rigorously considered by all studies, especially those published before the 2000s. In several cases, Indigenous identity was not specified in the studies or was not used as an inclusion criterion. For consistency, we intentionally excluded studies where the Indigenous community was not living in their country of origin, such as when cultural adaptation was done for immigrants or refugees in another country. In future research, these factors could be considered – particularly, the geographical distribution of Indigenous peoples in identified studies. Moreover, since there are many undertakings of research with Indigenous on mental health assessment, updated searches should be done to support new innovation in culturally safe tools.

## Conclusion

Revision of linguistic equivalence and cultural relevance of items, culturally safe administration procedures, and participatory research were key features of improving cultural sensitivity of psychometric measures for depressive symptoms among global Indigenous populations. Participatory qualitative research and methods are appropriate for conducting cultural adaptation of instruments most suitable for the community, but a careful, shared needs assessment is necessary to prioritize resources. Across all studies, increasing cultural safety by translation, changes to content, or modes of administration protocol were found to be empowering and healing. Indigenous reported that the most acceptable scales were culturally safe and able to be integrated into their health services. Use of visual scales and auditory materials are promising in terms of acceptability of the measurement. Finally, empowerment – focused approaches and tools as alternatives to symptom measures should actively be sought for provision of high-quality and culturally sensitive mental health services.

**Open peer review.** To view the open peer review materials for this article, please visit http://doi.org/10.1017/gmh.2023.75.

**Supplementary material.** The supplementary material for this article can be found at https://doi.org/10.1017/gmh.2023.75.

**Data availability statement.** All additional data is available upon request from authors.

**Acknowledgements.** We would like to thank Ms. Andrea Quaiattini from the McGill University Library for her assistance with our comprehensive systematic search procedure. We also express our gratitude for Indigenous co-authors in other publications and some other community members in Montreal and Kahnawà:ke for introducing us to some basic concepts of their view of health and wellbeing as opposed to symptom measurement.

**Author contribution.** L.G.C., V.N. and O.L. were involved in the conceptualization of the review. Project management was the responsibility of L.G.C. and O.L. Data screening and data analysis were conducted by all authors. All findings were validated by L.G.C. The original draft of the manuscript was written by L.G.C., M.Y., Q.S., and O.L. was reviewed and edited by L.G.C., M.Y., O.L. All co-authors reviewed the final version of the article. Funding was acquired by L.G.C. and O.L.

**Financial support.** O.L. was funded by grants from FSISSS (#8400958), CIHR (#426678), FRSQ (#252872 and #265693O), the Réseau Québécois sur le suicide, les troubles de l'humeur et les troubles associés, and the Strategic Research Council (SRC) established within the Academy of Finland (#352700). L.G.C. had financial support from the Réseau universitaire intégré de santé et services sociaux (RUISSS) McGill and CIHR grant #430331. Q.S. had funding from McGill University's Healthy Brains Healthy Lives fellowship (Q.S.). The funders had no role in study design or in later collection and data analysis.

**Competing interest.** The authors declare none.

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
