## [Reviewer Report]

Please find attached our review “The methods of improving cultural sensitivity of depression scales for use among Indigenous populations - A systematic scoping review” for consideration in the journal Global Mental Health. Culturally competent care is needed to address major depression amongst Indigenous communities, as rates of depression and suicide have commonly been found to be significantly higher amongst Indigenous in comparison to the general population. We present a systematic scoping review of processes where standard depression scales that have been culturally adapted for use among Indigenous populations worldwide. Here, we aimed to describe the processes and main features of adaptation methods and approaches.

---

## [Reviewer Report]

Thank you for asking me to read this interesting scoping review examining methods to improve the cultural sensitivity of Indigenous populations. This is important work as draws together the range of approaches used to adapt scales for use by Indigenous people, a practice which is increasing in focus by researchers and necessary to prepare health services to deliver care to Indigenous people. I believe this to be a potentially important paper, however, I provide some comments for your consideration.

Introduction

1. Some of these claims could be strengthened with references, for instance:

- Sentence three, can we be specific about examples of marginalisation? (starting “As a result of colonization…) and back it up by literature.

- Final sentence para 1, (starting “access and quality is commonly not equal…), can we define and reference this statement?

Aims

2. There is some inconsistency in the aims as reported throughout the paper, e.g., between the impact statement ( stated as “describing the concepts and methods use), abstract (“process and features of methods) and article (“availability of measures, target pops and geographic distribution of the scales, methods and process of adaptations). I suggest reviewing these for consistency to clarify the purpose for the reader.

3. I suggest reviewing the aims in the paper. From reading the results it appears that aim 1 (availability) is addressed in section 3.2. I’m not clear the difference between aims 1 (availability) and 2 (target populations and geo distribution). Perhaps these could be revised and simplified. The bulk of the results appear to be focused on aim 3 (“methods), section 3.3 onwards. (More closely linking the aims with the results will enhance compliance with PRISMA item 18).

Methods

4. The search is now over 2 years out of date. Perhaps the authors could consider updating this search or justifying why this is appropriate.

5. The authors have appropriately reported using the PRISMA ScR guidelines for the review. I was unable to review the search strategy (as is included in the guideline) as it was not provided with the manuscript, however it was noted that it would be provided as a supp table.

Results

6. I suggest considering the framing of communities to recognise their skills and strengths throughout the paper. For instance, Pg 9 line 120 identifies ‘the community and experts’, which could be interpreted to indicate the community are not experts, whereas the community holds a range of skills and knowledge that the research experts lack. In this instance, re-framing the ‘experts’ as ‘university based researchers or clinicians’ (as is accurate) may address this issue. This should be considered throughout.

7. PHQ-16 (ref 28) and PHQ-9 (ref 27) are the same tool, i.e., the PHQ-9 was adapted in ref 27, resulting in the aPHQ-9, which was then tested for validity, etc in ref 28. I suggest reviewing the representation of the results table to reflect this process.

Discussion

8. 4.4 I find it unusual to include additional results in the discussion that are not responding to an aim or result (sentence starting “Beyond the main objective of this review, we also identified some complementary projects..). When considering my comments above about the aims, I suggest considering including this information in the aim, result.

9. 4.4 Additional referencing in this section would improve clarity – currently, it is not clear to the reader which paper are been illuded to in this section and specific information about this idea.

10. As it currently reads, I found it difficult to follow how these conclusions were drawn. How did we ascertain if co-produced tools minimised stigma? (section: The studies showed that co-produced tools can minimize stigma associated with mental illness, increase acceptability of measurement among the population, have higher sensitivity to local emotional experiences, and improve identification and treatment of distress). How was this shown? By linking the aim, method with this idea may clarify this point.

11. 4.5 Consider reviewing and including a reference to demonstrate which literature the paper is referring to (line 284-5). I also wondered if the authors were aware of the Australian “Aboriginal Resilience and Recovery Questionnaire (ARRQ)”, by Graham Gee as this work may have relevance to this section.

---

## [Reviewer Report]

Thank you for the opportunity to review the manuscript The Methods of Improving Cultural Sensitivity of Depression Scales for Use Among Indigenous Populations – A Systematic Scoping Review.‘’ In this systematic scoping review, the authors seek to expand the literature on the strategies used in the cultural adaptation of standard depression scales for Indigenous populations worldwide.

Reviewer’s Comments:

Strengths

The manuscript will significantly contribute to the field with a high impact.

The manuscript is original and fills a gap in the literature.

The manuscript covers global content, including research inclusion, presentation of results, and discussion.

The findings will contribute to advancing knowledge in the field of psychometrics, psychiatry, and Indigenous mental health.

The authors convey their ideas and present their results in an organized and structured manner.

Areas for Improvement

Global content in the impact statement could be expanded.

Describing the reasons for the exclusion of the 3662 articles would be informative.

---

## [Reviewer Report]

Response to Question 1

For global reviews, how well does the review cover global content in the inclusion of research, presentation of results, and/or in the discussion and implications? And how could this be improved/expanded?

This scoping review investigates an important issue that is relevant to practitioners working with Indigenous populations and researchers investigating the prevalence of Western conceptualisation of depression and/or the impact of interventions. In doing this the manuscript contributes to a growing body of work that highlights the issues related to the use of screening tools that are designed using the Western biomedical paradigm and associated conceptualisation of depression.

However, in my view, in its current version the manuscript has several issues that should be attended to. Initially, the framing of the ‘problem’ does not take a strengths-based approach. Rather than focusing on Indigenous peoples as the problem, the authors should consider the benefits of framing the issue of cultural adaptation of tools developed using the Western biomedical paradigm as a recognition that the worldviews of Indigenous peoples globally are holistic being interrelated with and interconnected to their families, community, and lands. Consequently, they are different from the individualistic Western biomedical paradigm on which the screening tools examined in this scoping review are based on.

Second, the authors do not explicitly identify the root of the issue which is that Indigenous worldviews are different and therefore ‘dis-ease’ and the means of screening it/them need to be aligned with those worldviews.

Third, while the authors identified that they followed Tricco et al., (2018) scoping review checklist they did not report using an acceptable and repeatable method for their review. This scoping review would have been significantly enhanced using a published method. For example, Joanna Briggs Institute (https://jbi.global/scoping-review-network/resources) or Peters MD, Godfrey CM, Khalil H, McInerney P, Parker D, Soares CB. Guidance for conducting systematic scoping reviews. JBI Evidence Implementation. 2015 Sep 1;13(3):141-6, which includes using the framework Population, Concept Context (PCC) to frame inclusion criteria. Additionally, the Tricco et al., (2018) checklist (item number 4) identifies the need for “an explicit statement of the questions and objectives being addressed with reference to their key elements (e.g. population or participants, concepts and context) (p. 471).

Fourth, the review would be significantly enhanced if the authors systematically and explicitly defined the parameters of their subject of interest. For example, what do the authors mean by cultural relevance, linguistic equivalence, acceptability?

Fifth, the authors should review their discussion about Hackett et al., 2019 (beginning line 120). Hackett and colleagues (2019) (The Getting it Right Study) was a validation of Brown and colleagues previous work (2012, 2013, 2016) done to adapt the PHQ-9 with Australian Aboriginal people from the Central Desert area.

Lines 123 – 131 in the manuscript is taken from Hackett et al., 2019 (p. 24). As cited in the manuscript it misrepresents what the validation did. Here the authors should be referring to the work of Brown (2013).

Relevant references that support this adaptation are listed below.

Brown, A., Scales, U., Beever, W., Rickards, B., Rowley, K., & O’Dea, K. (2012). Exploring the expression of depression and distress in aboriginal men in central Australia: a qualitative study. BMC psychiatry, 12, 1-13.

Brown, A. D., Mentha, R., Rowley, K. G., Skinner, T., Davy, C., & O’Dea, K. (2013). Depression in Aboriginal men in central Australia: adaptation of the Patient Health Questionnaire 9. BMC psychiatry, 13, 1-10.

Brown, A., Mentha, R., Howard, M., Rowley, K., Reilly, R., Paquet, C., & O’Dea, K. (2016). Men, hearts and minds: developing and piloting culturally specific psychometric tools assessing psychosocial stress and depression in central Australian Aboriginal men. Social Psychiatry and Psychiatric Epidemiology, 51, 211-223.

Sixth, the inclusion of the word ‘global’ in the title would alert the reader to the fact that this scoping review included publications from across the globe.

Finally, the manuscript needs significant editing.

Recommendation – major revision

The following addresses specific aspects of the manuscript

Abstract

Both reliability and validity are necessary for all scales including adapted scales. Consequently, the sentence beginning “Therefore, cultural adaptation….” should include relevant validity, not just acceptability (face validity) and reliability.

The objective needs editing as the current sentence is fragmented. Perhaps it should read “…present findings of systematic review of ‘processes’ used to cross-culturally adapt depression scales for use with Indigenous peoples worldwide?”

Methods:

For standard translation do you mean language translation?

Conclusion:

Sentence reading “While for comparability, ….” also consider health equity.

Access to culturally safe measures that reflect local idioms and lexicon support equitable access to appropriate care and treatment.

Introduction.

Line 19 – also consider bias and equivalence of tools developed using the Western biomedical paradigm.

Lines 35 – 36 Why is the development of instruments based on DSM-5 important? What is different about Western and Indigenous expressions of low mood? Why does this matter?

Line 41 - The sentence beginning “While no consensus….” is true. However, Beaton and colleagues (2000), (Beaton DE, Bombardier C, Guillemin F, Ferraz MB. Guidelines for the process of cross-cultural adaptation of self-report measures. Spine. 2000; 25(24): 3186-91) (cited by 12494 according to Google Scholar; Arafat et al., (2016) (Arafat SMY, Chowdhury HR, Qusar M, Hafez MA. Cross cultural adaptation & psychometric validation of research instruments: A methodological review. J Behav Health. 2016; 5(3): 129-36) (cited by 259); and Sousa & Rojjansrirat (2011) (Sousa VD, Rojjanasrirat W. Translation, adaptation and validation of instruments or scales for use in cross‐cultural health care research: a clear and user‐friendly guideline. J Eval Clin Pract. 2011; 17(2): 268-74) (cited by 2291) have all published relevant guidelines (guides) that should be cited here as acknowledgment of the work that has already been done in this area.

Line 51 – The sentence beginning “The main objectives ….” Should be past tense.

Objective 1 – what do you mean by “track”?

Line 53 – are methods and process the same thing?

What are the research questions guiding this scoping review and what is its rationale?

The authors indicated that they had a problem in their study but are not explicit about what that problem was. The authors indicated that they are following the Tricco et al., 2018 scoping review checklist which identifies that a rationale (number 3 on the checklist) for the scoping review must be included.

Methods

The method contains insufficient information about ho the data was managed. Were citations uploaded to a citation manager? How did co-authors access the data?

Lines 67- 71 This needs a reference

Line 79 – Authors indicate that they calculated Kappa coefficients, but they are not reported in the manuscript.

Line 82 – inclusion criteria indicates that the authors included primary source studies. However, they included at least one protocol (Hackett et al., 2016) which is not a study.

Line 84 – Inclusion criteria 5 indicates that the authors included peer review articles. Why was peer review important? According to Munn et al., (2018) the purpose of a scoping review is “to identify knowledge gaps, scope a body of literature, clarify concepts or to investigate research conduct.” (p. 1). Scoping reviews map broad and available evidence which includes that provided in grey literature. Consequently, the authors need to justify why they have limited their inclusion criteria to peer-reviewed articles and indicate why a systematic review (which uses empirical evidence) was not a more appropriate method for their study.

Results:

Line 100 – 3845 articles is different to that recorded in the PRISMA flow chart (Fig 1).

Line 106 – Sentence beginning The Indigenous group …….”. Can the authors clarify what they mean by “scales were native to”?

Line 118 – Sentence beginning with “Further, three studies….”. The authors use the word methodology when it should be methods.

Lines 122- 123 The First Nations peoples of Australia are Aboriginal and Torres Strait Islanders. They are two culturally distinct groups. Torres Strait Islander people are of Melanesian descent.

Line 123 – Should Hackett et al., 2016 be included in the analysis? It is a protocol not a study.

Line 133 – What do the authors mean by “increase acceptance” and how does acceptance relate to the scale’s validity?

Lines 134 – 135 Relevant citations of articles that reported these characteristics would be helpful for the reader.

Lines 136 – 137 More information needed here including authors, location, relevant Indigenous population, and related scale.

Line 139 – is this just about comprehension (understanding)? Is it also about acceptability?

Line 140 – Refer to comment for Lines 122-23 for correct terminology for Australian First Nations peoples.

Line 159 – Citations needed here

Discussion

Line 211 – 212 I suggest that this information should be presented in the results section as it relates to Objective 1 (tracking)? In which countries are these scales most used?

Line 239 – There are more recent publications associated with the language translation of scales that should have been cited here. See comment for Line 41.

Line 242 – This sentence may be contrary to a previous comment made by the authors (Line 41) indicating that there are no guides or evaluation criteria.

Lines 241 –251 (Section 4.2) This section would benefit from citing relevant literature.

Lines 257 – Consider whether “high illiteracy rates” takes a strength-based approach (see first general comment). The use of deficit language such as this reinforces continuing impact of colonisation where use of dominant language is promoted above Indigenous languages.

Lines 264 - 277 (Section 4.4)

If this very important aspect about cross-cultural adaptation of scales is to be included in the manuscript. It needs to be included as a research question and objective of the scoping review as well as reported in the results section.

Line 269 – To assist the reader relevant studies should be cited here.

Line 309 – Is there a more recent reference than 1999?

Line 318 – Do you mean Indigenous peoples?

Line 322 – How do the review findings facilitate further research and equitable exchange of knowledge for resilience and supporting factors? This is very important when utilising a strengths-based approach that values the importance of connection to culture as supporting resilience.

References

Line 533 – Review journal name.

---

## [Reviewer Report]

Please find attached the revised article “The Methods of Improving Cultural Sensitivity of Depression Scales for Use Among Global Indigenous Populations - A Systematic Scoping Review”, which we kindly resubmitted for consideration in Cambridge Prisms: Global Mental Health. We feel very grateful for the comments from referees and believe edits have improved the scope of the review.

Yours,

Outi Linnaranta, MD, PhD

Finnish Institute for Health and Welfare, Finland

McGill University, QC, Canada

---

## [Reviewer Report]

Review of “The Methods of Improving Cultural Sensitivity of Depression

Scales for Use Among Global Indigenous Populations – A Systematic Scoping Review”

General comments

The authors have significantly strengthened this manuscript by incorporating reviewer feedback which is to be commended. However, there are opportunities to further strengthen thins manuscript by positing ‘up front’ the key reason why scales developed using the Western biomedical paradigm are inappropriate for Indigenous peoples globally. This issue is germane to the need for this scoping review. In my opinion positioning is not strong enough in the Introduction. Suggestions for strengthening this out outlined below.

In addition, there is an opportunity to use a strengths-based approach to your discussion about Indigenous peoples’ literacy levels.

It is also noted that all your examples come from the Canadian context (p. 13 line 262 – 263; p. 15 Lines 309 – 321). Given the global scope of this review this aspect could be further strengthened and more inclusive by including additional citations from other countries.

Finally, please note recommendations below in relation to Table 1. Specifically how you refer to the First Nations peoples of Australia and correctly reporting the work of Hackett and colleagues (2016; 2019) in relation to the validation of the aPHQ-9.

Specific comments

Pg 3 line 6 – 8 Review for written expression. What do you mean by roots?

Pg 3 line 10 – amongst Indigenous peoples?

Pg3 12 – 15 – This sentence would be strengthened by including decolonising psychology (Dudgeon et al., 2017; Dudgeon, 2020; Gone, 2021)as an approach to supporting the mental health of Indigenous peoples.

Line 18 – biased in what way? This point could be strengthened by including considerations of Indigenous worldviews of health and wellbeing. For example, Indigenous peoples holistic worldviews (Dudgeon et al., 2017; Gall et al., 2021; Weaver, 2002) that are inclusive of their connections to community and their lands which also makes screening tools developed using the individualistic Western bio-medical paradigm inappropriate. If this point was strengthened then it would link nicely with your additions related to tool acceptability (p10 -11) where, in line 205 (p.11) you speak to “insights into participants’ lived experiences and social, emotional and cultural wellbeing.”

P11. Line209 – Esler et al., (2007) reference is not needed twice.

P 13 line 262 – 263 Is there a more recent publication than Reading and Wien (2006)? Noting that Reading and Wein report Canadian information, it would also be good to include citations from other countries that are represented in your scoping review.

P13 line 263 – 266 This section does not take a strengths-based approach. Perhaps consider that Indigenous language traditions are mainly oral not written (Struthers & Peden-McAlpine, 2005) and the coloniser’s language is often a second or third language (+) of an Indigenous person and how that might impact their responses to questionnaires.

P15 Lines 309 – 321 Your inclusion of this section has strengthened your discussion. However, as this is a global scoping review perhaps use your Canadian reference as an example of the deleterious impact of colonisation. Other Indigenous peoples across the globe have experienced a variety of policies by their colonisers that have impacted their communities in similar ways. To be more inclusive, perhaps this point should also be noted.

P 18 lines 398- 99. Consider adopting a strength-based approach to literacy in line with suggestions above (p. 13 lines 263- 266)

Page 28 – 32 across Table one

Is ethnic group the most appropriate title for column 9? Perhaps Indigenous group would be more appropriate.

It is noted that you refer to Australian Indigenous and Torres Strait Islanders separately and inconsistently across Table 1. For example, EPDSb, K5; KMMS etc.

Torres Strait Islanders are one of the two groups of First Nations peoples of Australia. Therefore, using the term Australian Indigenous is inclusive of Torres Strait Islanders. If you need to differentiate it is suggested that you use Australian Aboriginal and Torres Strait Islander Peoples. Please review https://aiatsis.gov.au/explore/australias-first-peoples

Page 31 line 24 aPHQ-9 (amended) column six should be amended to describe what the study (Hackett et al., 2019) did, not the intention of the protocol. For example, focus groups were not conducted with participants only questionnaires (the aPHQ-9 and acceptability) were administered. Phone interviews were only conducted with participants if face-to-face interviews to establish the criterion validity of the aPHQ-9 using the MINI were not available.

Interviews were conducted to determine the feasibility of administering the aPHQ-9 and staff perceptions of participating in the study. However, these findings are reported in a separate publication (Farnbach et al., 2019).

This is an example were the inclusion of a protocol which is not a research study becomes issue in a systematic review as the intentions outlined in a protocol may not eventuate in the actual study as is the case with here.

References

Dudgeon, P., Bray, A., D’Costa, B., & Walker, R. (2017). Decolonising psychology: Validating social and emotional wellbeing. Australian Psychologist, 52(4), 316-325.

Dudgeon, P., Darlaston-Jones, D.; Alexi, J. (2020). Decolonising psychology: Self-determination and social and emotional well-being 1. In Routledge handbook of critical indigenous studies (pp. 100-113). Routledge.

Farnbach, S., Gee, G., Eades, A.-M., Evans, J. R., Fernando, J., Hammond, B., Simms, M., DeMasi, K., Glozier, N., Brown, A., Hackett, M. L., & Getting it Right, I. (2019). Process evaluation of the Getting it Right study and acceptability and feasibility of screening for depression with the aPHQ-9. BMC PUBLIC HEALTH, 19(1), 1270. https://doi.org/https://dx.doi.org/10.1186/s12889-019-7569-4

Gall, A., Anderson, K., Howard, K., Diaz, A., King, A., Willing, E., Connolly, M., Lindsay, D., & Garvey, G. (2021). Wellbeing of Indigenous Peoples in Canada, Aotearoa (New Zealand) and the United States: A Systematic Review. International Journal of Environmental Research and Public Health, 18(11). https://doi.org/10.3390/ijerph18115832

Gone, J. P. (2021). Decolonization as methodological innovation in counseling psychology: Method, power, and process in reclaiming American Indian therapeutic traditions. Journal of Counseling Psychology, 68(3), 259.

Struthers, R., & Peden-McAlpine, C. (2005). Phenomenological research among Canadian and United States Indigenous populations: Oral tradition and quintessence of time. Qualitative Health Research, 15(9), 1264-1276.

Weaver, H. N. (2002). Perspectives on wellness: Journeys on the red road. J. Soc. & Soc. Welfare, 29, 5.

---

## [Reviewer Report]

I have carefully reviewed the revised manuscript titled “The Methods of Improving Cultural Sensitivity of Depression Scales for the Use Among Global Indigenous Populations - A Systematic Scoping Review” that you submitted for consideration in Global Mental Health. I would like to commend you and your co-authors for the thorough revisions made in response to the previous feedback. The changes have significantly improved the clarity, quality, and overall scientific rigor of the manuscript. Thank you for the opportunity to review your work, and I look forward to seeing this valuable contribution published in the near future.

---

## [Reviewer Report]

Please find attached a second revision of our article “The Methods of Improving Cultural Sensitivity of Depression Scales for Use Among Global Indigenous Populations - A Systematic Scoping Review”, which we respectfully resubmit for consideration.

Yours,

Outi Linnaranta

Medical Director, Mental Health Strategy, Finnish Institute for Health and Welfare, Helsinki, Finland